# The Presence and Localization of G-Quadruplex Forming Sequences in the Domain of Bacteria

**DOI:** 10.3390/molecules24091711

**Published:** 2019-05-02

**Authors:** Martin Bartas, Michaela Čutová, Václav Brázda, Patrik Kaura, Jiří Šťastný, Jan Kolomazník, Jan Coufal, Pratik Goswami, Jiří Červeň, Petr Pečinka

**Affiliations:** 1Department of Biology and Ecology/Institute of Environmental Technologies, Faculty of Science, University of Ostrava, 710 00 Ostrava, Czech Republic; dutartas@gmail.com (M.B.); jiri.cerven@osu.cz (J.Č.); 2Faculty of Chemistry, Brno University of Technology, Purkyňova 118, 612 00 Brno, Czech Republic; xcfricova@fch.vut.cz (M.Č.); vaclav@ibp.cz (V.B.); 3Institute of Biophysics, Academy of Sciences of the Czech Republic v.v.i., Královopolská 135, 612 65 Brno, Czech Republic; jac@ibp.cz (J.C.); pratikgoswami@ibp.cz (P.G.); 4Faculty of Mechanical Engineering, Brno University of Technology, Technicka 2896/2, 616 69 Brno, Czech Republic; 160702@vutbr.cz (P.K.); stastny@fme.vutbr.cz (J.Š.); 5Department of Informatics, Mendel University in Brno, Zemedelska 1665/1, 61300 Brno, Czech Republic; jan.kolomaznik@gmail.com

**Keywords:** G-quadruplex, bacteria, bioinformatics, deinococcus, G4Hunter

## Abstract

The role of local DNA structures in the regulation of basic cellular processes is an emerging field of research. Amongst local non-B DNA structures, the significance of G-quadruplexes was demonstrated in the last decade, and their presence and functional relevance has been demonstrated in many genomes, including humans. In this study, we analyzed the presence and locations of G-quadruplex-forming sequences by G4Hunter in all complete bacterial genomes available in the NCBI database. G-quadruplex-forming sequences were identified in all species, however the frequency differed significantly across evolutionary groups. The highest frequency of G-quadruplex forming sequences was detected in the subgroup Deinococcus-Thermus, and the lowest frequency in Thermotogae. G-quadruplex forming sequences are non-randomly distributed and are favored in various evolutionary groups. G-quadruplex-forming sequences are enriched in ncRNA segments followed by mRNAs. Analyses of surrounding sequences showed G-quadruplex-forming sequences around tRNA and regulatory sequences. These data point to the unique and non-random localization of G-quadruplex-forming sequences in bacterial genomes.

## 1. Introduction

The discovery of the B-DNA structure by Crick and Watson started a rapid growth in genetic and molecular biology research [1]. However, it is now clear that, apart from this well-known double helical DNA structure, other forms of secondary structure participate in various basic processes [2]. The presence of various local DNA structures including cruciforms [3], quadruplexes [4] and triplexes [5] has been demonstrated by various methodological approaches. For example, G-quadruplexes (G4) were studied by crystallography as far back as 1962 [6]. G4s are secondary structures formed by guanine rich sequences which are widespread in DNA and RNA [4]. The building block for a G4 is a guanine quartet formed by G:G Hoogsteen base pairs (Figure 1). G4 formation requires the presence of monovalent cations such as Na^+^ and K^+^ [7]. Formation of this structure regulates various processes including gene expression [8], protein translation [9] and proteolysis pathways [10] in both prokaryotes and eukaryotes. In human, G4s are formed in various regions including sub-telomeres, gene bodies and gene regulatory regions [11] and in telomere regions to suppress degradation and maintain genomic stability [12]. Formation of G4s in this region decreases telomerase activity and decreases the chances of cancer development [13,14]. In addition, the proto-oncogene *MYC* is bound by nucleolin in its hypersensitive region III and enhances G4 folding and suppresses *MYC* transcription [15]. Therefore, it has been suggested that anticancer therapy will be possible by targeting G4s [16,17,18]. Moreover, it has been demonstrated that G4-stabilizing ligands modulate gene transcription [11]. It is already known that clusters of G4-forming sequences induce gene expression and that they are distributed near promoters and 5′UTRs. Replication-dependent DNA damage evidenced by G4 ligands have also been discovered in tumor suppressor genes and oncogenes [19]. Another potential therapeutic option is to target G4-binding proteins. Many proteins are known to bind to G4s, including some proteins important in cancer [20,21]. Moreover, novel G4 binding proteins have been suggested, sharing the NIQI amino acid motif (RGRGR GRGGG SGGSG GRGRG) [22].

Due to the roles of G4s in regulating basic cellular processes, it is essential to identify the location of G4s in genomes. Several algorithms for detecting expected matching patterns for G4 formation are already described. The first algorithm [G_n_N_m_G_n_N_o_G_n_N_p_G_n_] was created by Balasubramanian and colleagues [24] and the second algorithm considering occurrence of repeating unit G_n_ (n ≥ 2) was created by the group of Maizels [25]. Nevertheless, these algorithms only produce binary (yes/no; match/no match) results, rather than the quantitative analyses that are mandatory for correlation with quadruplex strength metrics. G4Hunter was developed to overcome this limitation, in which G4 propensity is calculated depending on G richness and G skewness [26]. 

Bacterial genetic material is stored mostly in circular chromosomes and plasmids [27]. It was demonstrated that secondary structures in bacterial genomes are responsible for genomic stability [28]. In addition, G4 structures are more stable than double stranded DNA due to slower unfolding kinetics [29]. Nevertheless, fewer studies on role of G4 in bacterial survival and virulence have been carried out [30]. A comparative functional analysis by Pooja et al. revealed that open reading frame (ORF) formulated amino acids biosynthesis and signal transduction are restrained by/controlled by G4 DNA in prokaryotes [31]. There are have been many reports on the role of G4s in eukaryotes over many years [32], although advances in prokaryotic G4s are not fully elucidated [33]. 

The formation of an intramolecular G4 requires the presence of a loop sequence between the G-tracts [34] and the density of G4 therefore broadly correlates with GC content. The GC content in bacterial genomes varies remarkably, from 17% to 75% [35]. It was demonstrated that G4 forming sequences are enriched and biased around transcription start sites of genes in the order Deinococcales [36]. Another function of G-tracts is in sustaining and maintaining duplex stability at higher temperatures in thermophiles; for example, *Thermus aquaticus* has a GC content of 68% [37]. Interestingly, the soil bacterium *Paracoccus denitrificans* contains 494 G4-forming sequences, which play roles in digestion of NO_3_- through G4 formation upstream of *NasT* [38]. The presence of G4-forming motifs in genes *hsdS*, *recD*, and *pmrA* of *Streptococcus pneumoniae* participate in host–pathogen interactions [30]. Such observations show the significance role of G4 in bacteria and also in eukaryotic cell organelles such as chloroplasts and mitochondria with circular DNAs that originated from prokaryotic organisms. Several papers show the importance of local DNA structures in mitochondrial DNA including G4 using G4Hunter [26] and inverted repeats [39] using palindrome analyzer [40]. Similarly, cruciforms exist in various regulatory regions in chloroplast DNA [41]. 

The presence of G4 in bacteria remains poorly understood. In our study, we comprehensively analyzed the presence and locations of G4 in 1627 bacterial genomes using G4Hunter. These data bring more information about evolutionary changes of G4 frequency between phyla and provide evidence for the importance of G4 in prokaryotes.

## 2. Results

### 2.1. Variation in Frequency for G4-forming Sequences in Bacteria

We analyzed the occurrence of putative G4 sequences (PQS) by G4Hunter in all 1627 known bacterial genomes. The length of bacterial genomes in the dataset varies from 298 kbp to 20.20 Mbp. The GC content average is 50.44%, with minimum 20.2% for *Buchnera aphidicola* (Gammaproteobacteria) and maximum 74.7% for *Corynebacterium sphenisci* (Actinobacteria). Using standard values for G4Hunter algorithm—window size 25 and G4Hunter score above 1.2—we found 9,202,364 PQS in all 1547 bacteria with 1627 genomes (some bacteria have two genomes). The most abundant PQS are those with G4Hunter scores of 1.2–1.4 (97.9% of all PQS), much less abundant are PQS with G4Hunter scores 1.4–1.6 (1.96% of all PQS), followed by 1.6–1.8 (0.128% of all PQS) and 1.8–2.0 (0.0056% of all PQS) and the lowest number of PQS is above G4Hunter score 2 (0.0009% of all PQS). In general, a higher G4Hunter score means a higher probability of G4s forming inside the PQS [26]. A summary of all PQS found in ranges of G4Hunter score intervals and precomputed PQS frequencies per 1000 bp is shown in Table 1.

According to NCBI taxonomy classification, the fully sequenced organisms of Bacteria domain are divided into 18 groups (6 with 10 or more sequenced genomes) and 39 subgroups (14 with 10 or more sequenced genomes), as shown in the phylogenetic tree (Figure 2). For statistical analyses, we used only groups with 10 or more sequenced genomes (highlighted by colors).

The number of all analyzed sequences in individual phylogenetic categories, together with median genome length, shortest genome, longest genome, mean, minimal and maximal observed frequency of PQS per 1000 bp and total PQS counts are shown in Table 2. Five subgroups (Actinobacteria, Chloroflexi, Deinococcus-Thermus, Alphaproteobacteria and Betaproteobacteria) show >60% GC content. On the other side, three subgroups (Spirochaetia, Thermotogae and Tenericutes) show < 40% GC content.

Mean frequency for all bacterial genomes was 1.342 PQS per 1000 bp. The lowest mean frequency is for Thermotogae (0.395) and the highest for the PVC group (1.646), followed by Terrabacteria (1.601). On the subgroup level, the lowest mean frequency was found in Tenericutes (0.136) and the highest in Deinococcus-Thermus (6.626), followed by Actinobacteria (2.821). The very highest PQS frequency of 14.213 PQS/kbp was found in *Thermus oshimai JL-2* (subgroup Deinococcus-Thermus) and the lowest frequency (0.013 PQS/kbp) in *Lacinutrix venerupis* (subgroup Bacteroidetes/Chlorobi) containing only 40 PQS in its 31,923,99 bp genome (0.0125 PQS/kbp). Detailed statistical inter group and inter subgroup comparisons are depicted in Appendix A.

Detailed statistical characteristics for PQS frequencies (including mean, variance, and outliers) are depicted in boxplots for all inspected subgroups (Figure 3). 

We visualized the relationship between %GC content in genomes with the frequency of PQS (Figure 4). In general, PQS frequencies usually correlate with GC content, however there are many exceptions to this rule. Organisms with high PQS frequencies relative to their GC content (over 50% of the maximal observed PQS frequency, Figure 4) are highlighted in color; the whole figure is separated into smaller segments according to inspected G4Hunter score intervals. Nearly all of the 10 outliers belong to the group Terrabacteria, except *Spirochaeta thermophila* DSM 6578 (group Spirochaetes). From the Terrabacteria group, six outliers belong to the small subgroup Deinococcus-Thermus (*Thermus oshimai* JL-2, *Thermus brockianus*, *Thermus aquaticus* Y51MC23, *Thermus scotoductus* SA-01, *Marinithermus hydrothermalis* DSM 14884, and *Deinococcus puniceus*), two outliers belong to the subgroup Actinobacteria (*Verrucosispora maris* AB-18-032 and *Rubrobacter xylanophilus* DSM 9941) and one outlier comes from the subgroup Cynobacteria/Melainabacteria (*Microcystis aeruginosa* NIES-843).

### 2.2. Localization of PQS in Genomes

To evaluate the position of PQS in bacterial genomes, we downloaded the described “features” of all bacterial genomes and analyzed the presence of all PQS in each annotated sequences and its close proximity (100 bp before and after feature annotation). PQS frequencies around annotated genome sites are shown in Figure 5. The highest PQS frequencies are before and after transfer RNA (tRNA), then inside transfer-messenger RNA (tmRNA) and inside ribosomal (rRNA). The lowest PQS frequencies were noticed before and after sequence-tagged sites (STS), then after and before rRNA and after miscellaneous features. If we consider only “inside” regions of inspected features, the differences between features are much smaller than within “before” and “after” regions.

As shown in Figure 5, there is no straight pattern in PQS occurrence in all annotated sequences but, in some groups, there are certain PQS distributions. For example, inside rRNA, tmRNA, ncRNA, misc_features, genes and repeat regions, there is higher amount of PQS in annotated sequences, but these PQS are not frequently present in DNA situated before and after these annotated sequences. In contrast, there is almost the same distribution of PQS before, inside and after annotated sequences in tRNA and regulatory groups. 

## 3. Discussion

It has been demonstrated that G4s could be used as targets for therapy [42]. G4 ligands are suggested as a target in cancer [43] and show antiparasitic activity for *Trypanosoma brucei* binding to a G4 structure [44]. Therefore, it has been proposed that G4 sequences in bacterial genomes represent novel and promising targets for antimicrobial therapy [33], and dinuclear polypyridylruthenium(II) complexes are active against drug-resistant bacteria including Methicillin-resistant *Staphylococcus aureus* and Vancomycin-resistant *Enterococcus* [45,46]. Dinuclear ruthenium complexes are relatively well-characterized G4 DNA binding agents [47,48,49]. Interestingly, we found large numbers of PQS with G4Hunter scores greater than 1.8 in the cyanobacterium *Microcystis aeruginosa*. *Microcystis aeruginosa* is a ubiquitous cyanobacterium living in eutrophic fresh water, which produces harmful hepatotoxins and neurotoxins, and can cause economic loss and damage to the ecosystem [50]. Our analysis indicates that this organism contains unusual and perfectly repeated PQSs (for example, DNA repeat of (GGGGTGT)_58_). Therefore, we hypothesize that this organism could be very sensitive to treatment with specific G4 binding compounds to inhibit its growth, as a possible alternative to commonly used algicides (the human genome does not contain these GGGGTGT repetitions). On the other hand, the lowest mean frequency of PQS was found in Terrabacteria subgroup Tenericutes (0.136 PQS/kbp) with the lowest average GC content (28%). The subgroup Tenericutes includes the genus *Mycoplasma* with many pathogens of clinical importance. On the other hand, a G4 was found in the promoter region of *Mycobacterium tuberculosis* and G4 ligands inhibited *M. tuberculosis* growth in the low micromolar range [51]. Therefore, the presence of a G4 could be important not only in antiviral [42,52] but also in antibacterial therapy. According to a recent study by Ding et al., eukaryotic organisms have similar PQS frequencies of 0.3 PQS per 1000 bp, whereas prokaryote frequencies are more diverse [36]. Based on our analysis, prokaryote PQS frequencies span a range of 0.013 (*Lacinutrix venerupis*) to 14.213 (*Thermus oshimai JL-2*) PQS per 1000 bp. A similar observation was shown by Quadparser algorithm and leads to the hypothesis that thermophilic organisms are enriched with PSQs due to their living at high temperatures [36]. However, similar enrichment has been demonstrated also for organisms with resistance to other stress factors such as radioresistance [53,54], thus the direct correlation between temperature and G4 presence is not supported by these finding. Validation of the G4Hunter score was made based on biophysical measurements at room temperature [26], therefore the number of G4 sequences in thermophiles could be overestimated, especially for those sequences with G4Hunter scores close to 1.2. Moreover, the mostly thermophilic and hyperthermophilic bacteria in phylum Thermotogae strains has one of the lowest PSQ frequencies. Thus, it seems that Gram-negative thermophilic bacteria evolved according to G4 structures in a completely different way than Gram-positive thermophilic bacteria, and that correlation among thermophiles and G4s depends on the phylum. Contrary to the enrichment of PQS near transcriptional start sites (TSS), 5′-3′UTR sequences and coding regions in eukaryotes [36], our analyses showed the highest PQS frequencies inside tmRNA, ncRNA and rRNA regions in prokaryotes (Figure 5). tmRNAs play a key role in the so-called ribosome rescue process, if ribosomes cannot finish translation, e.g. due to lost stop codon in translated mRNA. The physiological role of ncRNAs in prokaryotes is not fully elucidated, although they are considered to be important regulators of pathogenic processes by controlling virulence gene expression in *Staphylococcus aureus* and *Vibrio cholerae* [55]. The comparison of PQS frequencies between different studies could be complicated due to various PQS thresholds and algorithms. In our study, we used the state-of-the-art algorithm, G4Hunter, developed by Mergny and colleagues. This algorithm takes into account G-richness and G-skewness and has been experimentally validated [26]. Moreover, the current G4Hunter web version allows easy analyses of multiple genomes [56] and our comprehensive analysis showed the broad variations of PQS frequencies and their locations in bacterial genomes.

## 4. Methods

### 4.1. Selection of DNA Sequences

The set of all complete bacterial genomic DNA sequences was downloaded from the Genome database of the National Center for Biotechnology Information [57]. We used for our analyses only completely assembly level and we have selected one genome (representative) for each species (Appendix A) to avoid non-complete sequences and duplications. In total, we analyzed the presence of G4 sequences in 1627 genomes from the domain Bacteria, representing 5886 Mbp.

### 4.2. Process of Analysis

We used the computational core of our DNA analyzer software written in Java [40]. For these analyses, we used the G4Hunter algorithm implementation [56]. Parameters for G4Hunter was set to “25” for window size and G4 score above 1.2. An example of a putative G4 sequence found using such search criteria is provided in Appendix A. The overall results for each species group contained a list of species with size of genomic DNA and number of putative G4 sequences found (Appendix A). These data were processed by Python jupyter using Pandas (contains statistical tools). Graphs were generated from the Pandas tables using “seaborn” graphical library.

### 4.3. Analysis of Putative G4 Sequences Around Annotated NCBI Features

We downloaded the feature tables from the NCBI database along with the genomic DNA sequences. Feature tables contain annotations of known features found in the DNA sequence. We analyzed the occurrence of G4-forming sequences inside and around (before and after) recorded features. Features were grouped by the name stated in the feature table file. From this analysis, we obtained a file with feature names and numbers of putative G4 forming sequences found inside and around features for each group of species analyzed. Search for putative G4 forming sequences took place in a predefined feature neighborhood (we used ±100 bp—this figure is important for calculation of putative G4-forming sequence frequencies in feature neighborhoods) and inside feature boundaries. We calculated the amount of all predicted putative G4-forming sequences in regions before, inside and after features. An example of categorizing a putative G4-forming sequence according to its overlap with a feature or feature’s neighborhood is shown in Appendix A. Further processing was performed in Microsoft Excel and the data are available as Appendix A.

### 4.4. Phylogenetic Tree Construction

Exact taxid IDs of all analyzed groups were obtained from Taxonomy Browser via NCBI Taxonomy Database [58], downloaded to phyloT: a tree generator (http://phylot.biobyte.de) and a phylogenetic tree was constructed using function “Visualize in iTOL” in Interactive Tree of Life environment [59]. The resulting tree is shown in Figure 2.

### 4.5. Statistical Analysis

Statistical evaluations of differences in G4-forming sequences in phylogenetic groups were made by Kruskal–Wallis test in STATISTICA, with *p*-value cut-off 0.05; data are available in Appendix A.

## 5. Conclusions

In this research, we analyzed the presence of PQS in bacterial genomes. PQS were identified in all species, but the number of PQS differ remarkably among individual subgroups, showing evolutionary adaptations connected with G4. While the highest frequency of PQS was detected in Gram-positive extremophiles Deinococcus-Thermus subgroup, the lowest PQS frequency was found in Gram-negative thermophilic bacteria in Bacteroidetes/Chlorobi subgroup. Thus, it seems that evolution of these subgroups was driven by different strategies. PQS are enriched in ncRNA segments followed by mRNAs; analyses of surrounding sequences showed PQS enrichment also around tRNA and regulatory sequences. These data point to the unique and non-random localization of PQS in bacterial genomes. 

## Figures and Tables

**Figure 1 molecules-24-01711-f001:**
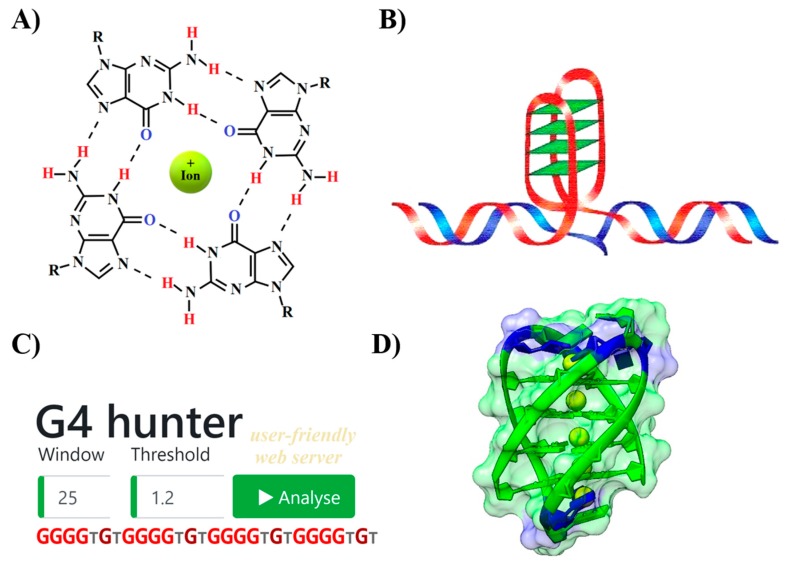
G-quadruplexes: (**A**) guanine tetrad stabilized by Hoogsten base pairing and positively charged central ion; (**B**) schematic drawing of intramolecular G4 structure arising from double stranded DNA; (**C**) G4Hunter, a new user-friendly web server for high throughput analyses of G4-forming sequences in DNA; and (**D**) 3D model of intramolecular antiparallel G4 formed from the sequence (5′-GGGGTGTGGGGTGT GGGGTGTGGGGTGT-3′) found in *Microcystis aeruginosa* built using 3D-NuS webserver [23].

**Figure 2 molecules-24-01711-f002:**
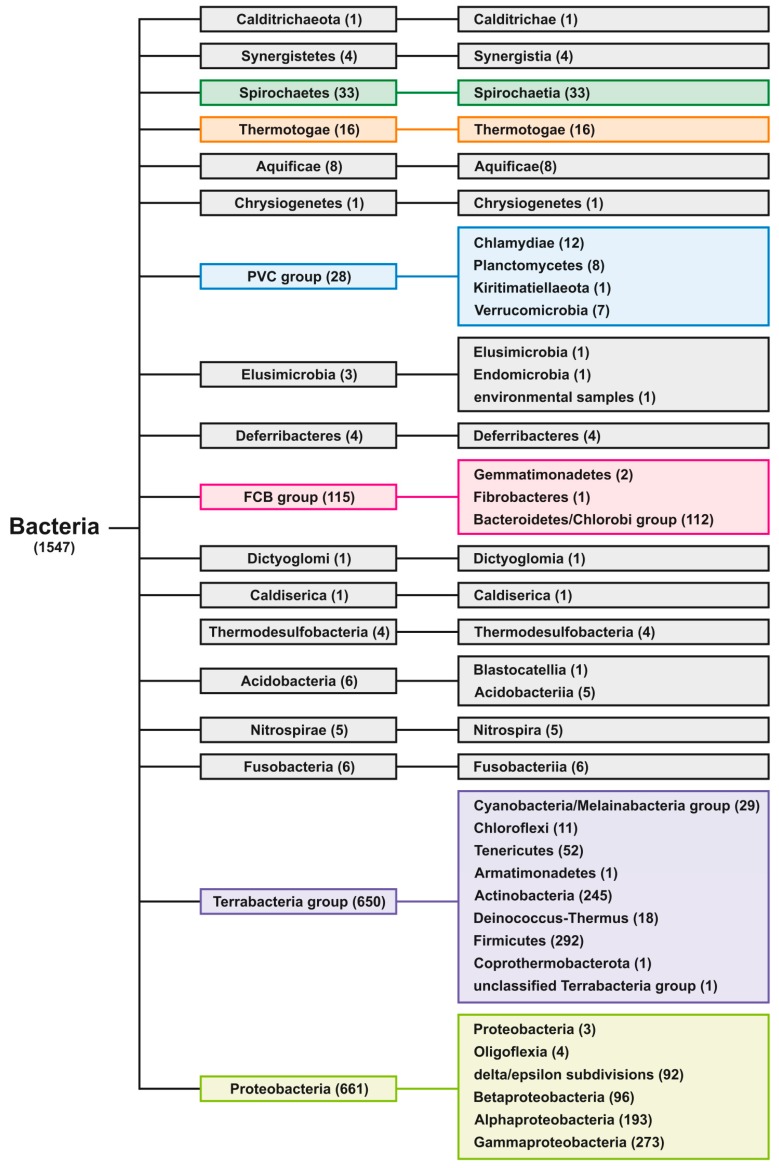
Phylogenetic tree of inspected Bacterial Groups and Subgroups.

**Figure 3 molecules-24-01711-f003:**
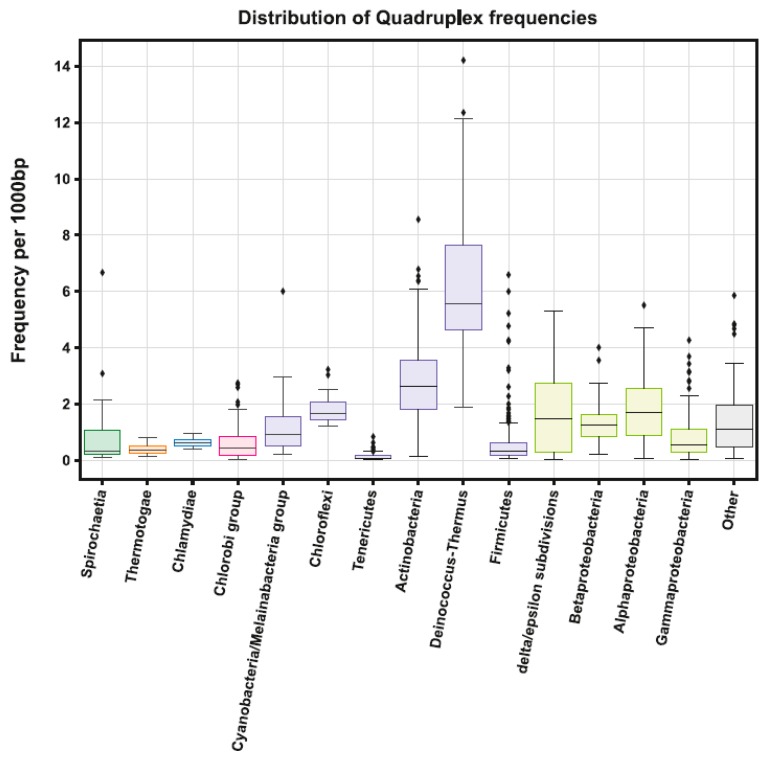
Frequencies of PQS in subgroups of the analyzed bacterial genomes. Data within boxes span the interquartile range and whiskers show the lowest and highest values within 1.5 interquartile range. Black diamonds denote outliers.

**Figure 4 molecules-24-01711-f004:**
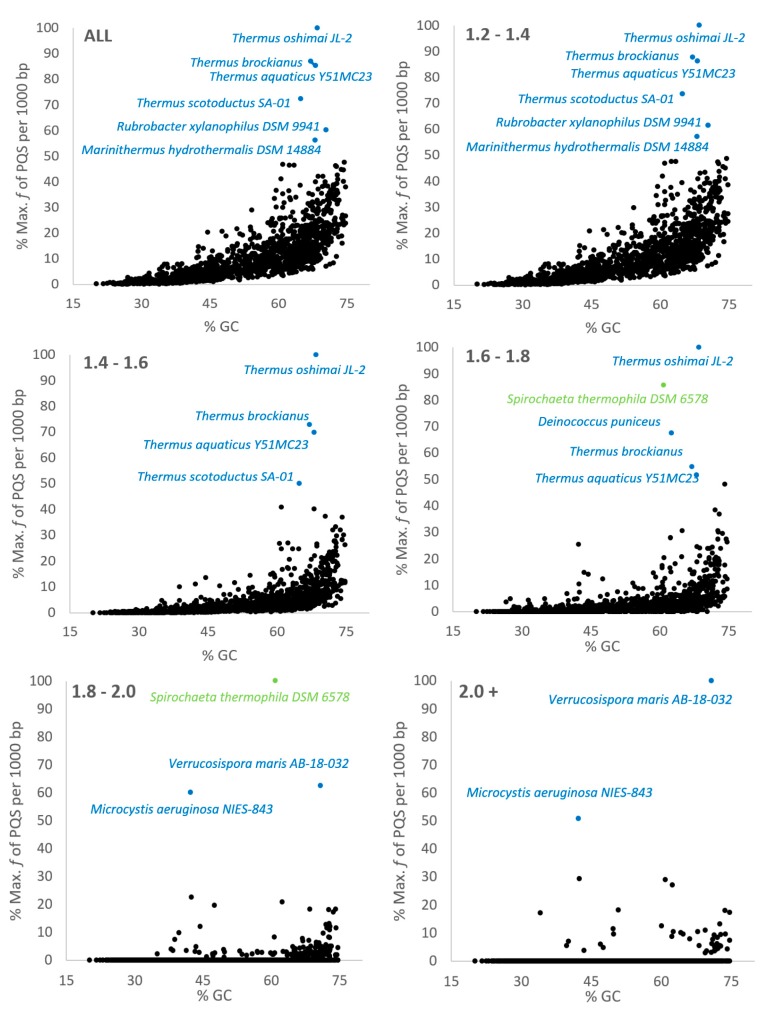
Relationship between observed frequency of PQS per 1000 bp and GC content in all analyzed prokaryotic sequences in various G4 Hunter score intervals. In each G4Hunter score interval miniplot, frequencies were normalized according to the highest observed frequency of PQS. Organisms with max. frequency per 1000 bp greater than 50% are described and highlighted in color.

**Figure 5 molecules-24-01711-f005:**
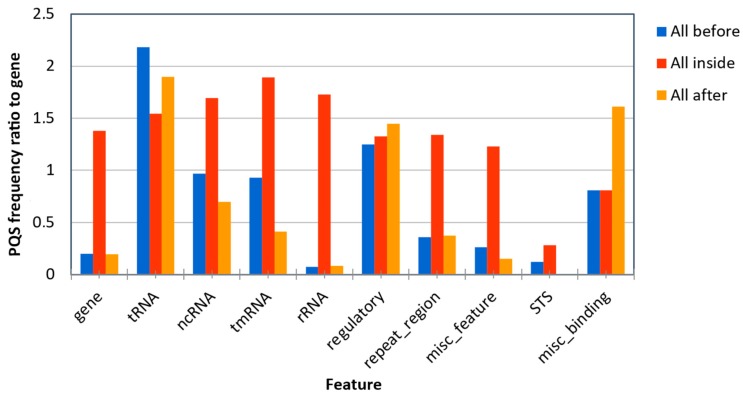
Differences in PQS frequency by DNA locus. The chart shows PQS frequencies according to “gene” annotation and other annotated locations from the NCBI database. We analyzed the frequencies of all PQS within (inside), before (100 bp) and after (100 bp) annotated locations.

**Table 1 molecules-24-01711-t001:** Total number of PQS and their resulting frequencies per 1000 bp in all 1547 representative bacteria, grouped by G4Hunter score. Frequency was computed by using total number of PQS in each category divided by total length of all analyzed sequences and multiplied by 1000.

Interval of G4Hunter Score	Number of PQS in Dataset	PQS Frequency per 1000 bp
1.2–1.4	9,009,593	1.315033
1.4–1.6	180,395	0.025058
1.6–1.8	11,779	0.00155
1.8–2.0	511	0.000055
2.0–more	86	0.000009

**Table 2 molecules-24-01711-t002:** Genomic sequences sizes, PQS frequencies and total counts. Seq (total number of sequences), Median (median length of sequences), Short (shortest sequence), Long (longest sequence), GC % (average GC content), PQS (total number of predicted PQS), Mean f (mean frequency of predicted PQS per 1000 bp), Min f (lowest frequency of predicted PQS per 1000 bp), Max f (highest frequency of predicted PQS per 1000 bp). Colors correspond to phylogenetic tree depiction.

**Domain**	**Seq**	**Median**	**Short**	**Long**	**GC%**	**PQS**	**Mean f**	**Min f**	**Max f**
Bacteria	1627	3,307,820	83,026	13,033,779	50.6	9,202,364	1.342	0.013	14.213
**Group**	**Seq**	**Median**	**Short**	**Long**	**GC%**	**PQS**	**Mean f**	**Min f**	**Max f**
Spirochaetes	38	2,646,038	277,655	4,653,970	39.7	87,109	0.809	0.079	6.668
Thermotogae	16	2,150,379	1,884,562	2,974,229	39.1	13,617	0.395	0.149	0.812
PVC group	28	2,917,407	1,041,170	9,629,675	50.7	198,358	1.646	0.388	4.802
FCB group	117	3,914,632	605,745	9,127,347	42.3	302,949	0.608	0.013	2.746
Terrabacteria	659	3,018,755	91,776	11,936,683	50.4	4,766,517	1.601	0.016	14.213
Proteobacteria	724	3,551,512	83,026	13,033,779	53.4	3,688,101	1.276	0.025	5.507
Other	45	2,157,835	1,012,010	6,237,577	44.3	145,713	1.103	0.062	5.855
**Subgroup**	**Seq**	**Median**	**Short**	**Long**	**GC%**	**PQS**	**Mean f**	**Min f**	**Max f**
Spirochaetia	38	2,646,038	277,655	4,653,970	39.7	87,109	0.809	0.079	6.668
Thermotogae	16	2,150,379	1,884,562	2,974,229	39.1	13,617	0.395	0.149	0.812
Chlamydiae	12	1,168,953	1,041,170	3,072,383	40.3	12,453	0.646	0.388	0.957
Bacteroidetes/Chlorobi	114	3,878,527	605,745	912,7347	41.9	282,516	0.585	0.013	2.746
Cyanobacteria/Melainab.	29	5,315,554	1,657,990	9,673,108	42.6	193,894	1.247	0.201	6.004
Chloroflexi	12	2,333,610	1,252,731	5,723,298	60	62,688	1.89	1.223	3.222
Tenericutes	52	981,001	564,395	1,877,792	28	6460	0.136	0.016	0.834
Actinobacteria	246	3,960,961	775,354	11,936,683	66.2	3,590,884	2.821	0.143	8.556
Deinococcus-Thermus	18	2,895,913	2,035,182	3,881,839	66.8	311,949	6.626	1.885	14.213
Firmicutes	298	2,835,823	91,776	8,739,048	40.8	579,740	0.56	0.064	6.587
delta/epsilon subdiv.	92	3,136,746	1,457,619	13,033,779	50	807,281	1.681	0.034	5.282
Betaproteobacteria	110	3,763,620	820,037	6,987,670	60.6	585,984	1.306	0.195	4.007
Alphaproteobacteria	213	3,424,964	83,026	9,207,384	61.5	126,134	1.764	0.051	5.507
Gammaproteobacteria	302	3,777,066	298,471	7,783,862	48.8	31,686	0.799	0.025	4.264
other	75	2,406,157	1,012,010	9,629,675	48.4	432,683	1.406	0.0616	5.855

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
