# Peer review of "The Presence and Localization of G-Quadruplex Forming Sequences in the Domain of Bacteria"

_molecules, 2019, doi:10.3390/molecules24091711_

Round 1
Reviewer 1 Report
The paper presents bioinformatic analyses of putative G-quadruplex forming sequences in the genomes of bacteria. One limitation of the paper is that it is a survey rather than an in-depth analysis providing key biological insight. Also, the found G-quadruplex forming sequences themselves are not provided. On the other hand one interesting aspect of the paper is having surveyed many bacterial organisms (1627 genomes). This paper thus present some new, though somewhat preliminary, data and will certainly call for future work.
Major points:
1) A discussion of the possible reason for frequency differences among organisms would be welcome. For example, the two organisms singled out in the abstract (Deinococcus Thermus and Thermotogae) are extremophiles. Could the fact that they have to function at higher temperature be a reason for requiring more stable G-quadruplexes? Can you attempt such correlations?
2) A related discussion to include is about the G4Hunter scoring and threshold. The validation of the G4Hunter score was made based on biophysical measurements at room temperature. It is thus possible that at the relevant temperatures for some organisms the score of 1.2 is not relevant because the G-quadruplexes would be too unstable, and thus the number of PQS is overestimated.
3) The introduction is hasty (some mechanisms such as those involved at telomeres have been much revised since and should be described with more nuances) and the references are not necessarily recent and sometimes not cited appropriately. Especially references 10-11-12-13 seem unrelated to the sentences introducing them. In the discussion, there are some sentences about "G4 ligands" which are so general that they are neither false nor true, because it does not take into account the variety of ligands. Even "dinculear ruthenium complexes" encompass a wide family of complexes and they don't necessarily all have the same behavior. This section is a bit too much of wishful thinking, because specific G4 ligands are still very rare.
4) The data analysis focused on the association between PQS frequency and DNA locus, and some discussion is about the potential to harness bacterial-specific PQS for antibiotic therapy. But that discussion would also have required a similar data analysis (same methodology) on eukaryotes and especially human genome. It would also have required an analysis and discussion of whether some specific sequences found in bacteria are unique to bacteria and/or among different bacteria.
Minor points:
- Abstract first line: "The importance of... is an emerging field" should be rephrased. The authors make abundant use of the expression "the importance of..." without explaining WHY what they are talking about is important. See also second sentence of abstract and line 92, for example.
- Intro first line, rephrase "started a boom".
- Figure 1, how the 3D model of panel D was obtained is not described. If it is just an elaborated drawing, delete it. Panel B suffices.
- line 65 "to overcome the limitations (of previous algorithms)..." -> specify which limitations and how, or to what extent, they are overcome by G4Hutner.
- line 82: NO3- with the "3" in subscript.
- in header of Table 1, specify how the frequency is defined. Check the number of significant figures in the frequencies.
- Figure 3 somewhat duplicates Table 2.
- line 184: "organisms with abnormally high PQS frequencies" --> what criteria is used to define normal and abnormal?
- line 206 delete "more" in "more smaller".
- Figure 5, typo in label "regulatory"
- reference 52 duplicates 21.
Author Response
Reviewer 1
Dear Reviewer,
We thank you very much for conducting the review of our manuscript, for your helpful comments and recommendations. These comments have helped us to improve our manuscript considerably. Detailed responses are in bold in this letter.
Major points:
1) A discussion of the possible reason for frequency differences among organisms would be welcome. For example, the two organisms singled out in the abstract (Deinococcus Thermus and Thermotogae) are extremophiles. Could the fact that they have to function at higher temperature be a reason for requiring more stable G-quadruplexes? Can you attempt such correlations?
Thank you very much for this suggestion. We have broadened Discussion section (page 10) as suggested, we have a strict deadline of 5 days to upload Minor revision of our article to Molecules, so unfortunately, we do not have time for detailed correlation analyses. On the one hand it seems that extremophiles in the phylum Deinococcus–Thermus (gram-positive, with the highest frequency of PSQs) are associated with their growth at high temperatures, on the other hand mostly thermophilic and hyperthermophilic bacteria in the phylum Thermotogae has one on the lowest PSQ frequency. Thus, it seems that gram-negative thermophilic bacteria evolved according to G4 structures in a completely different way than gram-positive thermophilic bacteria and that correlation among thermophiles and G-quadruplexes depends on the phylum.
2) A related discussion to include is about the G4Hunter scoring and threshold. The validation of the G4Hunter score was made based on biophysical measurements at room temperature. It is thus possible that at the relevant temperatures for some organisms the score of 1.2 is not relevant because the G-quadruplexes would be too unstable, and thus the number of PQS is overestimated.
Yes, obviously, it is possible, thanks for this comment, we commented on this aspect in the Discussion.
3) The introduction is hasty (some mechanisms such as those involved at telomeres have been much revised since and should be described with more nuances) and the references are not necessarily recent and sometimes not cited appropriately. Especially references 10-11-12-13 seem unrelated to the sentences introducing them. In the discussion, there are some sentences about "G4 ligands" which are so general that they are neither false nor true, because it does not take into account the variety of ligands. Even "dinculear ruthenium complexes" encompass a wide family of complexes and they don't necessarily all have the same behavior. This section is a bit too much of wishful thinking, because specific G4 ligands are still very rare.
Thanks, we have edited the Introduction and we have checked and added more recent and accurate references as suggested. On the other hand, we do not feel importance to comment deeply mechanisms involved in telomeres in the introduction because the bacterial genomes are usually circular and therefore this information is not directly relevant here.
4) The data analysis focused on the association between PQS frequency and DNA locus, and some discussion is about the potential to harness bacterial-specific PQS for antibiotic therapy. But that discussion would also have required a similar data analysis (same methodology) on eukaryotes and especially human genome. It would also have required an analysis and discussion of whether some specific sequences found in bacteria are unique to bacteria and/or among different bacteria.
Thanks for this comment. We agree, but these subsequent analyses will require much more work and computational resources and we are not able to provide the same analyses for eukaryotes within a minor review. To address your question, we have broadened the Discussion and used already published data with identical G4Hunter algorithm and we also compare our data with published results obtained by different algorithms. We have added this justification to the Discussion section. We also broadened the discussion about the PQS from Microcystis aureginosa, which is specific for this cyanobacteria (not found in Human), as checked by BLAT.
Minor points:
- Abstract first line: "The importance of... is an emerging field" should be rephrased. The authors make abundant use of the expression "the importance of..." without explaining WHY what they are talking about is important. See also second sentence of abstract and line 92, for example.
Thanks, we have rephrased these.
- Intro first line, rephrase "started a boom".
Thanks, we have rephrased.
- Figure 1, how the 3D model of panel D was obtained is not described. If it is just an elaborated drawing, delete it. Panel B suffices.
Thanks, it is our in silico model of this particular PQS, we have added appropriate description and quotation.
- line 65 "to overcome the limitations (of previous algorithms)..." -> specify which limitations and how, or to what extent, they are overcome by G4Hutner.
Thanks, we have specified some of them, however it is not the aim of our manuscript to compare different algorithms, the detailed comparison and advantages of G4Hunter algorithm is described in the paper: Bedrat, A.; Lacroix, L.; Mergny, J.L. Re-evaluation of G-quadruplex propensity with G4Hunter. Nucleic Acids Research 2016, 44, 1746–1759. We add this quotation to corresponding parts.
- line 82: NO3- with the "3" in subscript.
Thanks, we have fixed it.
- in header of Table 1, specify how the frequency is defined. Check the number of significant figures in the frequencies.
Thanks, we have fixed it.
- Figure 3 somewhat duplicates Table 2.
Yes, we agree that part of information is duplicated, however the graphical expression of these data in boxplots are more reader-friendly and serve mainly as a quick direct visual comparison of PQS frequencies between bacterial subgroups and Table 2 contains exact number, CG content etc. – so these information are not duplicated and are important for readers.
- line 184: "organisms with abnormally high PQS frequencies" --> what criteria is used to define normal and abnormal?
Thanks, we have rephrased it.
- line 206 delete "more" in "more smaller".
Thanks, we have deleted as suggested.
- Figure 5, typo in label "regulatory"
Thanks, we have fixed.
- reference 52 duplicates 21.
Thanks, we have checked and corrected references.
Yours sincerely,
Prof. Petr Pecinka
(on behalf of all authors)
Department of Biology and Ecology
Faculty of Science, University of Ostrava
petr.pecinka@osu.cz
+420 553 46 2318
Reviewer 2 Report
In this manuscript, Bartas M. et al. analyzed both the presence and locations of G-quadruplex-forming sequences in all the bacterial genomes available in the NCBI database (1627) by using the previously developed G4-Hunter algorithm.
The authors evaluated different species underlining that the frequency of G-quadruplex-forming sequences differed significantly across evolutionary groups, with the highest and lowest frequency respectively found in the Deinococcus-Thermus and in Thermotogae subgroups.
In addition, G-quadruplex-forming sequences were detected around tRNA and regulatory sequences of the bacterial genomes.
The G4-Hunter algorithm is not new: indeed, it has been developed some years ago by J.-L. Mergny and coworkers. Also, the use of this bioinformatic program has been already validated in different genomes.
The major merit of this work relies in the systematic analysis of the bacterial genomes, allowing a complete panorama of the putative G4 sequences in all groups and subgrups.
Overall, this study gives a valuable and original contribution, providing new evidences for the importance of G4 in prokaryotes The topic is attractive and of interest for the typical readership of Molecules.
The work is well organized and comprehensively described. In addition, the exploited methodology and data analysis may be interesting for a broad number of researchers working on the identification of G4 tracts in genomes in order to develop new potential strategies or drugs (as peculiar G4 ligands).
I would recommend its publication in Molecules provided that a few major revisions are carried out.
Major points:
In other published articles, after PQSs identification, the corresponding sequences were characterized by standard biophysical methods to determine the ability to effectively adopt G4 folding in vitro.
In the present version of the manuscript, the sequences found have not even be listed with the exception of the sentence reported in the discussion part: “Our analysis indicates that this organism contains unusual and perfectly repeated PQSs (for example DNA repeat of (GGGGTGT)15.”
At least, I suggest to introduce a new Table with the sequences found. However, including some preliminary data (for example CD spectra) of some of the sequences found would greatly enrich the importance of the presented work.
A list of minor points to be fixed is here enclosed:
1) In the introduction, when you refer to the importance of G-quadruplex structures, I will suggest to insert recent review articles on this topic.
2) Pag. 2, line 49: “G4 ligands” and not “G4 ligand”.
3) Pag. 2, lines 64-65: “To overcome limitations of these algorithms…” Please, clarify these limitations to the readers.
4) Pag. 3, line 86: “showed” in place of “show”.
5) Pag. 4, line 144: “In general, a higher G4Hunter score means a higher probability of G4s forming inside the PQS.” Please, insert a suitable reference for this statement.
6) Pag. 10, line 248: “This algorithm takes into account G-richness and G-skewness and has been experimentally validated.” Also this sentence requires a suitable citation.
7) Ref. 21 and ref. 52 are the same article: please correct! Also ref. 35 and 38 are the same.
8) Also, cite “J. Med. Chem. 2018, 61, 1231−1240” as example of G4 sequence identification in the genome of protozoan parasites.
Author Response
Reviewer 2
Dear Reviewer,
We thank you very much for conducting the review of our manuscript, for your helpful comments and recommendations. These comments have helped us to improve our manuscript considerably. Detailed responses are in bold in this letter.
Major points:
In other published articles, after PQSs identification, the corresponding sequences were characterized by standard biophysical methods to determine the ability to effectively adopt G4 folding in vitro.
In the present version of the manuscript, the sequences found have not even be listed with the exception of the sentence reported in the discussion part: “Our analysis indicates that this organism contains unusual and perfectly repeated PQSs (for example DNA repeat of (GGGGTGT)15.”
At least, I suggest to introduce a new Table with the sequences found. However, including some preliminary data (for example CD spectra) of some of the sequences found would greatly enrich the importance of the presented work.
Thank you for this comment, we will consider to do such biophysical evidences for selected PQS in the future. The aim of this article was to provide a comprehensive analysis of PQS occurrence in all sequenced bacterial genomes. Unfortunately, we are not able to enclose the sequences, because there were nearly 10 million of PQS found. On the other hand, our user-friendly webserver which implemented G4Hunter can be used by anyone for free, for particular PQS analysis in selected genome of interest. The dataset of analyzed sequences is too big to be enclosed and evaluate single PQS.
A list of minor points to be fixed is here enclosed:
1) In the introduction, when you refer to the importance of G-quadruplex structures, I will suggest to insert recent review articles on this topic.
Thank you we have checked and added more recent quotations.
2) Pag. 2, line 49: “G4 ligands” and not “G4 ligand”.
Thanks, we have fixed it.
3) Pag. 2, lines 64-65: “To overcome limitations of these algorithms…” Please, clarify these limitations to the readers.
Thanks, we have specified some of them, however it is not the aim of our manuscript to compare different algorithms, the detailed comparison and advantages of G4Hunter algorithm is described in the paper: Bedrat, A.; Lacroix, L.; Mergny, J.L. Re-evaluation of G-quadruplex propensity with G4Hunter. Nucleic Acids Research 2016, 44, 1746–1759. We add this quotation to corresponding parts.
4) Pag. 3, line 86: “showed” in place of “show”.
Thanks, we have changed as suggested.
5) Pag. 4, line 144: “In general, a higher G4Hunter score means a higher probability of G4s forming inside the PQS.” Please, insert a suitable reference for this statement.
Thanks, we have added an appropriate reference.
6) Pag. 10, line 248: “This algorithm takes into account G-richness and G-skewness and has been experimentally validated.” Also this sentence requires a suitable citation.
Thanks, we have added an appropriate reference.
7) Ref. 21 and ref. 52 are the same article: please correct! Also ref. 35 and 38 are the same.
Thanks, we have checked and corrected references.
8) Also, cite “J. Med. Chem. 2018, 61, 1231−1240” as example of G4 sequence identification in the genome of protozoan parasites.
Thanks, we have added this quotation.
Yours sincerely,
Prof. Petr Pecinka
(on behalf of all authors)
Department of Biology and Ecology
Faculty of Science, University of Ostrava
petr.pecinka@osu.cz
+420 553 46 2318
Reviewer 3 Report
In this study, the authors analyzed the presence and locations of G-quadruplex-forming sequences in all complete bacterial genomes, and suggested that they were unique and non-random localization. This is an interesting topic to clarify the importance of G4 in prokaryotes. The manuscript is clearly written and well analyzed. Therefore, the manuscript is acceptable for this journal.
Author Response
Reviewer 3
In this study, the authors analyzed the presence and locations of G-quadruplex-forming sequences in all complete bacterial genomes, and suggested that they were unique and non-random localization. This is an interesting topic to clarify the importance of G4 in prokaryotes. The manuscript is clearly written and well analyzed. Therefore, the manuscript is acceptable for this journal.
Dear Reviewer,
We thank you very much for conducting the review of our manuscript and appreciate your positive opinion about it.
Yours sincerely,
Prof. Petr Pecinka
(on behalf of all authors)
Department of Biology and Ecology
Faculty of Science, University of Ostrava
petr.pecinka@osu.cz
+420 553 46 2318